# Integrated Transcriptomics and Metabolomics Reveal the Mechanism of Alliin in Improving Hyperlipidemia

**DOI:** 10.3390/foods12183407

**Published:** 2023-09-13

**Authors:** Min Zhang, Xiaoying Zou, Yixuan Du, Zhuangguang Pan, Fangqing He, Yuanming Sun, Meiying Li

**Affiliations:** 1Guangdong Provincial Key Lab of Food Safety and Quality, South China Agricultural University, Guangzhou 510642, China; spxyzm@stu.scau.edu.cn (M.Z.); zouxiaoying97@stu.scau.edu.cn (X.Z.); duyixuan@stu.scau.edu.cn (Y.D.); hn20213141055@stu.scau.edu.cn (Z.P.); ymsun@scau.edu.cn (Y.S.); 2College of Food, South China Agricultural University, Guangzhou 510642, China

**Keywords:** alliin, transcriptomics, metabolomics, hyperlipidemia

## Abstract

This research aims to assess the anti-hyperlipidemia effects of alliin in vivo and its potential mechanisms through transcriptomics and metabolomics analysis. A hyperlipidemia mode was established in C57BL/6 mice fed a high-fat diet, and the related physiological parameters of the animals were recorded. Serum TC and MDA in livers significantly decreased by 12.34% and 29.59%, respectively, and SOD and CAT in livers significantly increased by 40.64% and 39.05%, respectively, after high doses of alliin interventions. In total, 148 significantly different genes, particularly *Cel*, *Sqle*, *Myc*, and *Ugt1a2,* were revealed for their potential roles in HFD-induced alliin, mainly through steroid biosynthesis, triglyceride metabolism, drug metabolism–cytochrome P450, and the PI3K–Akt signaling pathway, according to transcriptomics analysis. Metabolomics results revealed 18 significantly different metabolites between the alliin group and HFD group, which were classified as carboxylic acids, such as N-undecanoylglycine, adipic acid, D-pantothenic acid, cyprodenate, and pivagabine. We found pantothenic acid played a vital role and was effective through pantothenic acid and CoA biosynthesis metabolism. The “steroid biosynthesis pathway” was identified as the most significant metabolic pathway by integrated transcriptomics and metabolomics analysis. This work offered a theoretical framework for the mechanism of alliin lipid lowering in the future. The development and utilization of alliin will be a viable strategy to improve the health status of people with hyperlipidemia, suggesting prospective market opportunities.

## 1. Introduction

Garlic and its active components are being used more frequently in human diets and nutritional care since their beneficial effects are widely acknowledged. Early studies reviewed the active ingredients and functions of garlic and found that garlic’s multiple notable biological activities were primarily attributed to its organosulfur compounds [1,2]. S-Allyl-L-cysteine sulfoxide, known as alliin, constitutes the highest content of sulfur-containing compounds in fresh garlic, accounting for 6 to 14 mg/g and making up more than 90% of all sulfur-containing compounds [3]. Alliin is classified as the standard active component of garlic in several pharmacopeias, such as those of the United States. When it comes to the active ingredient of garlic, diallyl thiosulfinate, commonly known as allicin, is more extensively recognized than alliin. In fact, alliin is the most important naturally occurring sulfur-containing compound in fresh garlic. Alliin reacts with cysteine sulfoxide lyase to produce allicin. Allicin is highly unstable and readily decomposes into a range of different sulfur-containing chemicals [4]. The majority of allicin in commercially available products is chemically synthesized [5]. However, alliin is naturally abundant and stable, making it a suitable candidate for preservation. Additionally, alliin has excellent physicochemical properties, such as being odorless, non-volatile, and soluble in water [6,7]. Based on the findings of this study, alliin is better suited as a raw ingredient for healthy food and medicines than allicin. However, research on alliin is substantially less comprehensive than that on allicin and its degradation products. This is mainly due to early misunderstandings about the natural active components of garlic and the fact that certain pharmacopoeias and regulations set degradation products of allicin as the standard active ingredient of garlic.

According to the World Health Organization (WHO), about one quarter of adults worldwide suffer from hyperlipidemia. The overall prevalence of dyslipidemia in Chinese adults is 40.4% and is expected to reach 9.2 million cases from 2010 to 2030 [8]. Hyperlipidemia has become significantly more prevalent in recent years [9,10]. However, common lipid-lowering medications have some adverse effects and harm patients’ physical and mental health [11,12]. Traditional botanicals are being used by an increasing number of patients [13]. Garlic and its active components have been shown to lower total cholesterol (TC) and triglyceride (TG) levels in animals. However, most of the earlier lipid-lowering studies on garlic and its preparations only briefly assessed their lipid-lowering efficacy [14,15,16,17], whereas a few concentrated on intestinal flora [3] or certain lipid-lowering mechanisms, such as antioxidant effects and inhibition of lipid anabolic enzyme activity [18,19,20]. Several studies indicated that alliin, the most significant sulfur-containing component in garlic, had a good lipid-lowering impact [3,15]. Alliin efficiently inhibited 1,3-DCP-induced lipid synthesis in HepG2 cells, according to Jing et al., by activating the AMPK-SREBPs signaling pathway [21]. Animal studies by Itokawa et al. provided more proof that alliin might lower blood cholesterol, and enhance superoxide dismutase (SOD), catalase (CAT), and malondialdehyde (MDA), levels in the liver [22,23]. These studies focused solely on physiological indices, such as blood lipids and antioxidant activities, without in-depth studies on the lipid lowering mechanism. The potential molecular mechanism of alliin lipid-lowering was investigated by transcriptomics and metabolomics in our study. Our findings indicate a theoretical framework for alliin lipid-lowering research. Alliin-related products will be a viable strategy to improve the health status of people with hyperlipidemia, in the future suggesting prospective market opportunities.

## 2. Materials and Methods

### 2.1. Chemicals and Materials

Alliin (purity > 98% *w*/*w*) was provided by Haibo Biotechnology Co. (Xi’an, China). Serum biochemistry kits for determining the levels of SOD, MDA, and CAT were supplied by Nanjing Jiancheng Biological Engineering Institute Co., Ltd. (Nanjing, China). A basic diet (GD450B) and a high-fat diet (GD60) were purchased from Guangdong Medical Laboratory Animal Center (Guangzhou, China).

### 2.2. Animal Experimental Design and Sample Collection

The research procedure was authorized and overseen by the Center for Experimental Animals at South China Agricultural University (Guangzhou, China) (approval no: SYXK (Cantonese) 2019-0136). All studies were conducted in compliance with the Guidelines for Animal Protection and Utilization of South China Agricultural University in Guangzhou, China. Forty 5-week-old male C57BL/6 mice with initial weights of 18 to 22 g were provided by the Animal Center of Southern Medical University (Guangzhou, China) (Laboratory animal production license number: SCXK (Cantonese) 2021-0041). The animals were kept in the SPF barrier system of the South China Agricultural University’s Laboratory Animal Center, which maintained indoor temperatures of (23 ± 2 °C), relative humidity (60 ± 10)%, and a 12 h light/dark cycle. Animals had unlimited access to food and water.

After one week of adaptive feeding, all mice were randomly divided into four groups, with ten mice in each group (n = 10), namely, the control group (CON), the high-fat group (HFD), the low-alliin group (30 mg/kg) (LS), and the high-alliin group (120 mg/kg) (HS). The LS and HS groups, respectively, received 30 mg/kg and 120 mg/kg of alliin through intragastric administration, whereas the corresponding amounts of sterile water were given to the control and high-fat groups. The gavage of 0.1 mL per 100 g of body weight was estimated for mice. The control group was fed a basic diet (GD450B), and the remaining groups were fed a high-fat diet (GD60). The exact nutritional composition of the experimental diets is shown in Appendix A. During the experiment, the food intakes of each group of animals were recorded regularly every day, and the mice’s bodies were weighed once a week.

After eight weeks of feeding, the mice underwent a 12 h fast and dehydration period. In brief, the mice were anesthetized using isoflurane, blood samples were collected from their eyes, and the mice were sacrificed upon being fully anesthetized. The upper layer of blood was then collected after 15 min of centrifugation at 3500 rpm. The liver, perirenal fat, and epididymis fat were quickly separated and weighed. To observe the liver’s histology, a piece of the liver lobe was removed and fixed in 4% (*w*/*v*) paraformaldehyde. After being quickly frozen in liquid nitrogen, all samples were stored at −80 °C.

### 2.3. Blood Lipids Detection

The obtained blood was transferred to a microfuge tube after euthanizing the animals and centrifuged at 4 °C and 3500 rpm for 15 min to separate the plasma from blood cells. The obtained plasma was stored in a freezer until use. Serum TC, TG, LDL-c, and HDL-c were measured using an automatic biochemical analyzer (Mindray BS 380, Beckmancoulter, Guangzhou, China), which was based on the principle that blood samples are processed chemically and optically in order to quantify the lipid content of the blood. Throughout the experiment, we adhered to the necessary quality control procedures recommended by the machine manufacturer. All data processing and statistical analyses were performed using standard protocols.

### 2.4. SOD, CAT, and MDA Detection in Mice Liver

A commercial assay kit (Nanjing Building Ltd., Nanjing, China) was utilized to measure the activities of the three enzymes SOD, CAT, and MDA in the livers of the mice. The experiments were meticulously conducted following the manufacturer’s instructions. SOD activity was assessed by measuring the dismutation of superoxide radicals generated by xanthine oxidase and hypoxanthine, using Cu/Zn-SOD as the standard. The concentration of CAT enzyme in the sample was calculated by observing the elimination of hydrogen peroxide. The concentration of MDA was obtained by condensation with thiobarbituric acid (TBA), followed by a colorimetrical analysis of the resulting red product, which exhibited a maximal absorption peak at 532 nm. 

### 2.5. Histological Analysis

Liver tissues were fixed with 4% (*w*/*v*) paraformaldehyde. The fixed tissues were processed sequentially for paraffin embedding and then cut into sections, which were dyed with hematoxylin and eosin. Stained areas were observed by a positive-position fluorescence microscope (ZEISS, Axio Imager D2, Shanghai, China).

### 2.6. Transcriptomics Analysis

Total RNA was extracted from the tissue using Trizol^®^ reagent according to the manufacturer’s instructions, and genomic DNA was removed using DNase I. Then, RNA quality was determined by a 2100 Bioanalyzer and quantified using the ND-2000 (NanoDrop Technologies, Thermo Fisher Scientific, Milford, MA, USA). Only high-quality RNA samples (OD260/280 = 1.8~2.2, OD260/230 ≥ 2.0, RIN ≥ 0, RINOD260/28 ≥ 1 μg) were used to construct the sequencing library. The cDNA was enriched by PCR, and DNA-clean beads were used to screen 200–300 bp bands. After quantification by TBS380 (Picogreen), the library was sequenced using the Illumina HiSeq xten/NovaSeq 6000 sequencing platform for high-throughput sequencing with a read length of PE150 [24,25]. Genes referenced the Mus_musculus library for sequence alignment (http://asia.ensembl.org/Mus_musculus/Info/Index, accessed on 2 February 2023). Genes with |log2FC| = 2 and *p*-value < 0.05 were significantly differentially expressed genes (DEGs) [26]. DESeq2 was the program used for differential analysis. The data were analyzed on the online platform of Majorbio Cloud Platform (www.majorbio.com, accessed on 2 February 2023).

### 2.7. Determination of mRNA Expression Leve ls by qRT-PCR in Liver

Liver tissues were homogenized, and total RNA was isolated using a total RNA kit. A Nanodrop ultramicro spectrophotometer (Thermo Fisher Scientific, Waltham, MA, USA) was used to measure the total RNA concentration. Total RNA was reverse transcribed using HifairIII 1st Strand cDNA Synthesis SuperMix for qPCR to form the cDNA templates. The qPCR primers were designed and synthesized by Ruibiotech (Beijing, China), and the forward and reverse sequences are shown in Appendix A. The quantitative real-time PCR reaction mixture was set up using Heiff qPCR SYBR Green Master Mix (Low Rox). The values of the target genes in the liver were normalized to GAPDH. The relative expression levels were shown as (2^−ΔΔCt^) relative to the control group.

### 2.8. Metabolomics Analysis

Liver samples from the CON, HFD, and HS groups were selected for untargeted metabolomics analysis. Metabolomics analysis was measured as previously described by Park and Zou with some modifications [27,28]. Briefly, 50 mg of liver was mixed with pre-cooled 200 μL of acetonitrile and 100 μL of cold water. The mixture was vortexed for 1 min, ground at 35 Hz for 4 min, and sonicated in an ice bath for 5 min. The sample was subjected to three additional rounds of grinding and sonication before being stored overnight at −80 °C. The top layer was collected after centrifugation at 12,000 rpm for 15 min, and 5 μL of the sample was injected into the Acquity UPLC BEH C18 column (2.1 mm × 100 mm, 1.7 μm, Waters, Milford, MA, USA). The mobile phase, consisting of acetonitrile (phase A) and water with 0.1% (*v*/*v*) formic acid (phase B), was used in a gradient elution scheme with a flow rate of 0.15 mL/min. The gradient elution schemes were set at 95–85% (*v*/*v*) B for 0–3 min, 85–70% (*v*/*v*) B for 3–11 min, 70–75% (*v*/*v*) B for 11–15 min, 50–10% (*v*/*v*) B for 15–21 min, and 10–95% (*v*/*v*) B for 21–22 min. The mass spectrometry data were processed using Compound Discover 4.0 software along with the HMDB database and the mzCloud database (www.mzCloud.org, accessed on 14 May 2023) for qualitative analysis. Furthermore, we used MetaboAnalyst software to screen and analyze metabolites with |log2FC| ≥ 1.5 & *p*-value < 0.05.

### 2.9. Statistical Analysis

All results are shown as mean ± standard deviation (SD). With Duncan’s test *p* < 0.05, a way analysis of variance (ANOVA) was conducted.

## 3. Results

### 3.1. Effects of Alliin on Body Weight, Feed Intake, Fat Weight, and Liver Weight in Mice

Throughout the experiment, mice in the control group were in good condition, drinking and eating normally, with clean and shiny hair, no abnormal behavior, and no illnesses or deaths. The animals in the high-fat group were in good condition. They urinated and defecated normally, and their hair gradually became lusterless and greasy. After each dose of alliin intervention, the mice moved normally without abnormal behavior. There was no significant difference compared to the other groups.

After a feeding period of 8 weeks, body weight, feed intake, fat weight, and liver weight in each group were determined, as shown in Figure 1. The results showed no significant difference in body weight, feed intake, and fat weight in the HFD group compared to the alliin group. But, liver weight was significantly reduced in hyperlipidemia mice after alliin intervention. After receiving a high-fat meal, the mice’s body weight and fat weight both increased, but there was no discernible reduction after receiving alliin therapy (Figure 1A,C,D). Food intake in mice fed a high-fat diet after alliin intervention was a non-significant difference (*p* > 0.05), indicating that energy balance was not affected (Figure 1B). Moreover, liver weight increased significantly in the HFD group compared to the CON group while decreasing significantly through alliin intake (*p* < 0.01) (Figure 1E).

### 3.2. Effects of Alliin on Blood Lipids

The effect of alliin on blood lipids is displayed in Figure 2, including TC, TG, HDL-C, and LDL-C. Serum TC significantly increased by 31.05% (*p* < 0.001) while significantly decreasing by 12.34% and 16.58% (*p* < 0.01 or *p* < 0.05) after high and low doses of alliin interventions, respectively. Serum TG and LDL-C both increased between the HFD and CON groups but did not change significantly following alliin consumption. Serum HDL-C increased significantly between the HFD and CON groups while significantly dropping following alliin intervention.

### 3.3. Effects of Alliin on SOD, CAT, and MDA in Mice Liver

Following the consumption of a high-fat diet, SOD and CAT in the livers were significantly decreased 72.22% and 40.35%, respectively (*p* < 0.001). However, SOD was significantly elevated by 40.64% and 27.85% (*p* < 0.001) after high and low doses of alliin interventions, respectively, and CAT was significantly elevated by 39.05% and 31.99% (*p* < 0.001), respectively (Figure 3A,B). MDA in the livers significantly increased by 27.73% (*p* < 0.05) in the HFD group compared with the CON group and decreased by 29.59% and 23.52% (*p* < 0.05) after intake of high and low doses of alliin, respectively (Figure 3C). The results indicated that alliin was effective in ameliorating the liver stress injury induced by a high-fat diet in mice.

### 3.4. Effects of Alliin on Liver Histomorphology

Alliin affected the cell morphologies of liver in mice with hyperlipidemia (Figure 4). Hepatic cells in the CON group were neatly arranged and homogenous compared to the HFD group, which became bigger and contained more lipid vacuoles. The cell morphologies were considerably improved close to the normal cells by alliin supplementation. The result implied that alliin supplementation somewhat relieved the liver damage.

### 3.5. Transcriptomics Analysis

To investigate the anti-hyperlipidemia mechanism of alliin, we analyzed the differentially expressed genes of mice livers in the control, HFD, and high-alliin groups. The common genes were 9556, and the unique genes were 299, 326, and 127, respectively, among the three groups for Venn analysis (Figure 5A). There were large between-group differences in PCA analysis (Figure 5B). Great intra-group correlation was revealed according to the gene clustering analysis (Figure 5C). Furthermore, there were 514 significant differences genes among the three groups using p < 0.05 and FC ≥ 2 as the standards, as indicated in Appendix A. The HFD group had 405 differentially expressed genes compared to the control group, of which 189 were upregulated, and 216 were downregulated, indicating that a high-fat diet intervention resulted in a significant change in gene expression (Figure 5D). The alliin intake resulted that 61 genes were down-regulated, and 87 genes were up-regulated (Figure 5E). These results suggested that alliin intake regulated gene expression in high-fat-diet-fed mice.

Regulation of gene expression controls cell structure and function. GO and KEGG analyses were employed to further explore the function of genes. GO enrichment analysis showed that alliin primarily regulated some processes, such as phosphofructokinase activity, cholesterol metabolic process, cAMP-mediated signaling, steroid metabolism, and triglyceride metabolic processing, etc. (*p* < 0.05) (Figure 6A). The KEGG enrichment pathway included steroid biosynthesis, HIF-1 signaling, triglyceride metabolism, the drug metabolism–cytochrome P450 pathway, etc. (*p* < 0.05) (Figure 6B). We further explored related genes by drawing an enriched chord diagram of the KEGG pathway. *Gp5*, *Ugt1a2*, *Col4a4*, *Myc*, etc., were significantly up-regulated, whereas *Cel*, *Cele2a*, *Sqle*, *Camk2b*, etc., were significantly down-regulated after alliin intake in the HFD group. Most of these genes were related to lipid metabolism, such as steroid biosynthesis, drug metabolism–cytochrome P450, and the PI3K–Akt signaling pathway. In comparison to the control group, the HFD group increased the expression of *Cel* and *Sqle* in the steroid biosynthesis pathway (Figure 6C). The alliin supplement reduced the expression level of *Cel* and *Sqle* in the steroid biosynthesis pathway, whereas it increased the expression level of *Ugt1a2* and *Myc* in the drug metabolism–cytochrome P450 pathway and PI3K–Akt signaling pathway (Figure 6D). These findings implied that alliin intervention affected gene and signaling pathways related to lipid metabolism in high-fat-diet-fed mice.

### 3.6. Confirmation of Differential Expression of Lipid Metabolism Genes by qRT-PCR Analysis

Alliin intervention affected the expression of genes related to lipid metabolism in high-fat-diet-fed mice according to transcriptomics results. Thus, we examined the expression level of *Cel*, *Sqle*, *Myc*, and *Ugt1a2* related to lipid metabolism, through qRT-PCR (Figure 7). The expression level of *Cel* and *Sqle* were noticeably elevated in the HFD group compared to the CON group (*p* < 0.001 and *p* < 0.05) (Figure 7A,B). The expression levels of *Cel* and *Sqle* significantly reduced after alliin treatment (*p* < 0.001, *p* < 0.01, and *p* < 0.05) (Figure 7A,B). The levels of *Myc* and *Ugt1a2* in the HFD group were lower than those in the control group, but not noticeably different, whereas alliin intake markedly raised the levels of *Myc* and *Ugt1a2* (Figure 7C,D).

### 3.7. Metabolomics Analysis

Samples from individual groups were largely strongly clustered together in the PCA Figure (Figure 8A). The PLS-DA plot displayed large differences across groups (Figure 8B). Nine metabolites were upregulated, and seven metabolites were downregulated in the HFD group compared with the CON group, illustrating that a high-fat diet caused the distinguished metabolites profiles (Figure 8C). A total of 10 metabolites were raised, and 8 metabolites declined in the alliin group compared with the HFD group (Figure 8D). There were 18 significantly different metabolites between the alliin group and HFD group in Appendix A, if *p* < 0.05 and FC ≥ 1.5. These metabolites were classified as carboxylic acid, such as N-Undecanoylglycine, Adipic acid, D-pantothenic acid, Cyprodenate, and Pivagabine. KEGG analysis mainly contained pantothenic acid and CoA biosynthesis, and drug metabolism-cytochrome P450 pathway (Figure 8E).

### 3.8. An Association Analysis of Differential Genes and Metabolites in Mice Liver

We conducted an association analysis of these distinctly differential genes and metabolites between the alliin group and HFD group to clarify the metabolic pathways that genes and metabolites are involved in together (Figure 9). These results revealed that 51 metabolic pathways significantly changed after alliin supplement (Appendix A). The “steroid biosynthesis pathway” was the most significant metabolic pathway, which came at the top among the 51 significantly enriched signal pathways. Other metabolic pathways have also undergone significant changes, such as Terpenoid backbone biosynthesis, Glycerolipid metabolism, Glycerophospholipid metabolism, Steroid hormone biosynthesis, Alanine, aspartate and glutamate metabolism, Arginine biosynthesis, Drug metabolism-other enzymes, Tyrosine metabolism, etc.

## 4. Discussion

People have utilized garlic as a natural health remedy for hundreds of years [29,30,31]. The research will be a prominent topic on the use of alliin as adjuvants to traditional disease treatment in the future when garlic and its extracts are widely recognized. Alliin, the most significant sulfur-containing component in whole garlic, plays an active regulatory role in lipid metabolism. In this study, alliin consumption modulated blood lipids, ameliorated liver damage, and reduced oxidative stress levels, consistent with previous studies [4,14,21,32].

Our study elucidated the molecular mechanism of alliin’s anti-hyperlipidemic effects by integrating transcriptomics and metabolomics data. Transcriptomics analysis revealed 148 genes significantly regulated by alliin therapy, particularly, *Cel*, *Sqle*, *Myc*, and *Ugt1a2*. *Cel* encodes carboxyl ester lipase (CEL), an enzyme crucial for hydrolyzing cholesterol esters and triglycerides. The absence of *Cel* interfered with normal dietary fat uptake and increased susceptibility to diet-induced obesity in experiments using *Cel* knockout mice. The results showed that *Cel* regulated lipid metabolism and dietary fat absorption [33]. Previous population experiments have reported that a single nucleotide polymorphism mutation of *Cel* may potentially affect blood lipid metabolism and lead to the development of cardiovascular diseases [34]. *Cel* may play a significant role in the development of hyperlipidemia, consistent with our findings. *Sqle*, which encoded squalene epoxidase, was another gene that was upregulated by alliin treatment and had been shown to play a key role in cholesterol biosynthesis. Studies have reported a correlation between *Sqle* mutations and lipid abnormalities [35]. Furthermore, a study investigating the regulation of the *SREBP-1c* pathway in hyperlipidemia found that the expression level of *Sqle* was associated with cholesterol metabolism [36]. These results suggest a potential involvement of *Sqle* in hyperlipidemia development. *Myc* was a gene regulator in various cellular processes, including lipid metabolism [37]. A positive correlation was between *Myc* overexpression and hyperlipidemia. Mice fed with a high-fat diet overexpressing *Myc* in liver tissue showed abnormal glucose and lipid metabolism, whereas inhibiting *Myc* led to decreased blood triglyceride levels in rats with hyperlipidemia [38]. *Ugt1a2*, an enzyme responsible for metabolizing various hormones and drugs, was also modulated by alliin treatment. This is the first time *Ugt1a2* has been implicated in alliin’s anti-hyperlipidemia mechanism, and our results suggest that this enzyme may also play a role in hyperlipidemia development. Our qPCR results demonstrated that the expression levels of *Myc* and *Ugt1a2* were significantly upregulated, whereas *Cel* and *Sqle* were significantly downregulated after alliin intervention in high-fat mice, consistent with that previously observed our RNA-Seq data. Our findings confirmed the potential roles of these four genes in alliin’s anti-hyperlipidemic effects and extended the understanding of the lipid-lowering effects of alliin treatment. Meanwhile, these genes were involved in steroid biosynthesis, triglyceride metabolism, drug metabolism–cytochrome P450, and the PI3K–Akt signaling pathway, which were significantly enriched. Studies showed that alliin effectively attenuated 1,3-DCP-induced lipogenesis by regulating the expression of the fatty acid synthesis-related gene *SREBP-1c* [20]. Animal studies also showed that inhibition of ACC and *SREBP-1c* expression could reduce renal fatty acid levels and triacylglycerol levels [39,40,41]. Earlier studies have reported that the metabolic enzyme cytochrome P450 breaks down a variety of endogenous cholesterol and polyunsaturated fatty acids and is part of signaling pathways controlling cell cycles, apoptosis, invasion, and adhesion [42,43]. The PI3K/AKT signaling pathway has been shown to alleviate damage to the aortic intima, reduce collagen fiber content in the aorta, decrease blood lipid levels, and prevent oxidative stress, inflammatory response, and death in aortic vascular cells [44]. Drug metabolism–cytochrome P450 and the PI3K–Akt signaling pathway are connected to lipid metabolism.

Changes in genes and signaling pathways could affect metabolic function. Therefore, we further investigated liver metabolite changes in mice after alliin intervention. Metabolomics results found significant differences in metabolic pathways involved in pantothenic acid and CoA biosynthesis and drug metabolism–cytochrome P450 pathway. The pantothenic acid and CoA biosynthesis processes were essential for regulating numerous metabolic processes, such as energy generation, fatty acid and amino acid catabolism, and synthesis of fatty acids, phospholipids, sphingolipids, cholesterol, and steroid hormones [45,46]. The drug metabolism–cytochrome P450 pathway was involved in the metabolism of drugs and endogenous compounds, including some lipid-metabolism-associated substances such as cholesterol and estrogen in the liver [47]. These two pathways were closely related to lipid metabolism, further validating transcriptome results. The significantly different metabolites, including N-undecanoylglycine, adipic acid, and D-pantothenic acid, were related to hyperlipidemia. Among them, pantothenic acid [48] was the key precursor for the biosynthesis of coenzyme A (CoA), an essential cofactor involved in the synthesis of steroids. Research found that urine samples from patients with genetic metabolic disorders showed abnormal accumulation of several N-acyl amino acids, including N-undecanoylglycine, suggesting that N-acylglycines may be associated with certain inherited metabolic disorders [49]. Adipic acid was converted into acetyl-CoA through the adipic acid cycle and participated in the β-oxidation of fatty acids, which was closely related to the fatty acid metabolism pathway [50]. Therefore, N-undecanoylglycine, adipic acid, and D-pantothenic acid could potentially affect liver metabolism in mice, leading to the development of metabolic diseases. Earlier research showed that alliin was mostly absorbed from the small intestine into the blood in the intact form and partly converted to allyl sulfenic acid, pyruvic acid and ammonia [51]. We performed targeted metabolic quantification experiments on alliin, diallyl disulfide, and diallyl trisulfide in the liver of mice after alliin intervention, to further clarify that alliin itself or its metabolites or intermediates improve lipid metabolism. However, none of these three sulfur-containing compounds were detected, which may have been related to the fact that alliin has little hepatic first-pass effect and is rapidly metabolized in the body without easy accumulation. Alliin belonged to the fast elimination class of drugs and was basically eliminated from plasma after 120 min [52,53]. This deserves further study by future generations.

The “steroid biosynthesis pathway” was identified as the most significant metabolic pathway by integrated transcriptomics and metabolomics analysis, ranking first among all metabolic pathways. Cholesterol metabolism is essential to maintain cellular and bodily functions, and abnormal cholesterol metabolism is closely linked to hyperlipidemia [54]. Our results showed that the expression level of *Cel* and *Sqle* were both significantly reduced in the steroid pathway. *Cel* is a carboxyl ester lipase that catalyzes the hydrolysis of substrates such as cholesteryl esters, lysophospholipids, diacylglycerol, and triacylglycerol [34]. *Sqle* was recognized as the second-limiting enzyme for cholesterol biosynthesis downstream of *HMGCR* [55]. A study showed that cholesterol metabolism-associated RNA functions are implicated in the mechanisms of *Sqle* [56]. These genes are closely associated with lipid metabolism [8,57]. This suggests that alliin regulates lipid metabolism by regulating the expression level of *Cel* and *Sqle* in the steroid biosynthesis pathway.

In conclusion, results that included serum biochemistry and liver histology demonstrated that alliin improved high-fat-diet-induced hyperlipidemia. *Cel*, *Sqle*, *Myc*, and *Ugt1a2* played crucial roles in alliin’s anti-hyperlipidemic effects through the steroid biosynthesis pathway, triglyceride metabolism, drug metabolism–cytochrome P450 pathway and PI3K–Akt signaling pathway, with steroid biosynthesis being most significant. Details are shown in Figure 10. Our research elucidated alliin’s anti-hyperlipidemic mechanism through transcriptomics and metabolomics analysis, laying the foundation for further studies.

## 5. Conclusions

In conclusion, the results that included serum biochemistry, liver oxidative stress, and liver histology demonstrated that alliin improved high-fat diet-induced hyperlipidemia by regulating lipid metabolism-related pathways, especially the steroid biosynthesis pathway. Moreover, this study also elucidated the pharmacological mechanism and potential therapeutic effect of alliin through the integrative application of hepatic transcriptomics and metabolomics. Generally, alliin is odorless, soluble in water, stable, and simple to obtain from garlic. We can maximize the retention of alliin by inactivating the enzyme in garlic. Hence, the development and utilization of alliin products is a promising way to enhance the health status of people with hyperlipidemia, hinting at potential market prospects.

## Figures and Tables

**Figure 1 foods-12-03407-f001:**
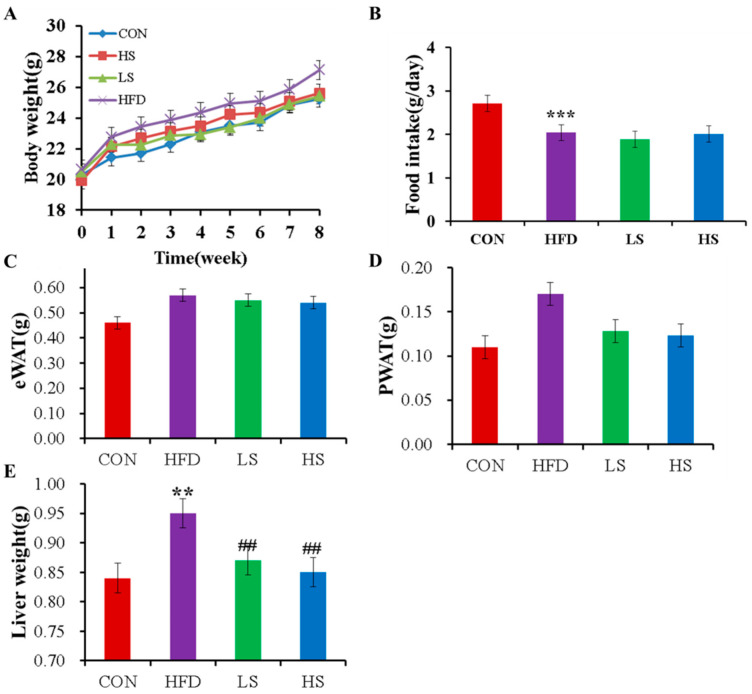
Effect of alliin on body weight (**A**), food intake (**B**), epididymal fat (**C**), perirenal fat (**D**), and liver weight (**E**) in mice. CON: the control group, HFD: the high-fat group, LS: the low-alliin group (30 mg/kg), and HS: the high-alliin group (120 mg/kg). ** (*p* < 0.01), and *** (*p* < 0.001) compared the CON group with the HFD group; ## (*p* < 0.01) compared the HFD group with the LS or HS group.

**Figure 2 foods-12-03407-f002:**
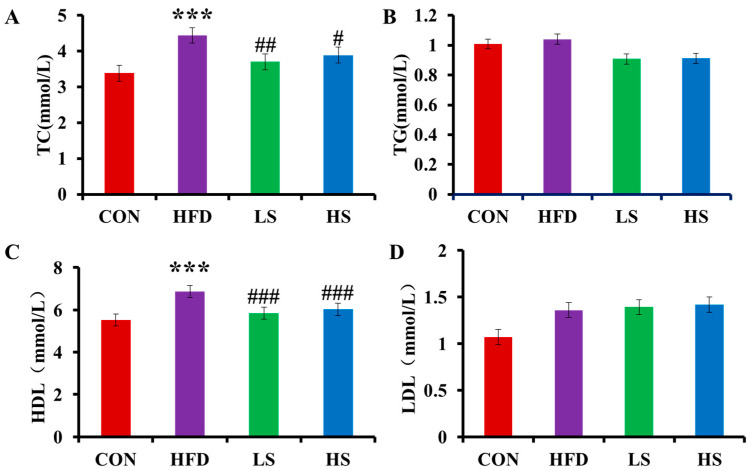
Effect of alliin on serum TC (**A**), TG (**B**), HDL-C (**C**), and LDL-C (**D**). *** (*p* < 0.001) compared the CON group with the HFD group; # (*p* < 0.05), ## (*p* < 0.01), and ### (*p* < 0.001) compared the HFD group with the LS or HS group.

**Figure 3 foods-12-03407-f003:**
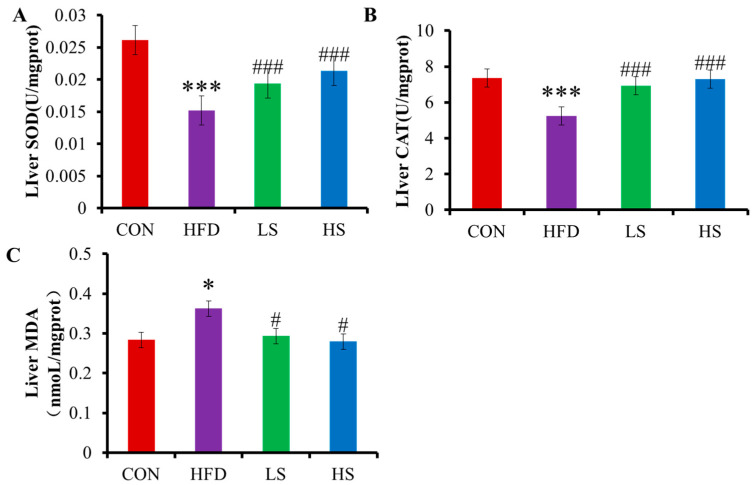
Effect of alliin on liver SOD (**A**), CAT (**B**), and MDA (**C**). * (*p* < 0.05), and *** (*p* < 0.001) compared the CON group with the HFD group; # (*p* < 0.05), and ### (*p* < 0.001) compared the HFD group with the LS or HS group.

**Figure 4 foods-12-03407-f004:**
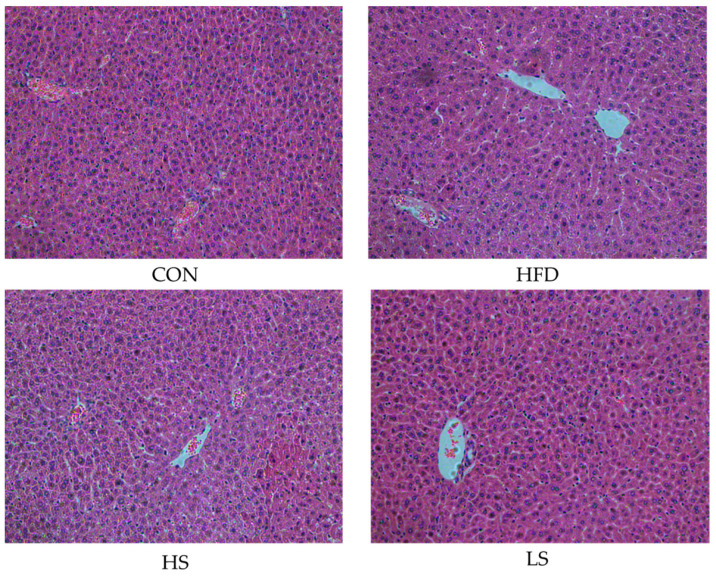
H&E staining map of liver (200×).

**Figure 5 foods-12-03407-f005:**
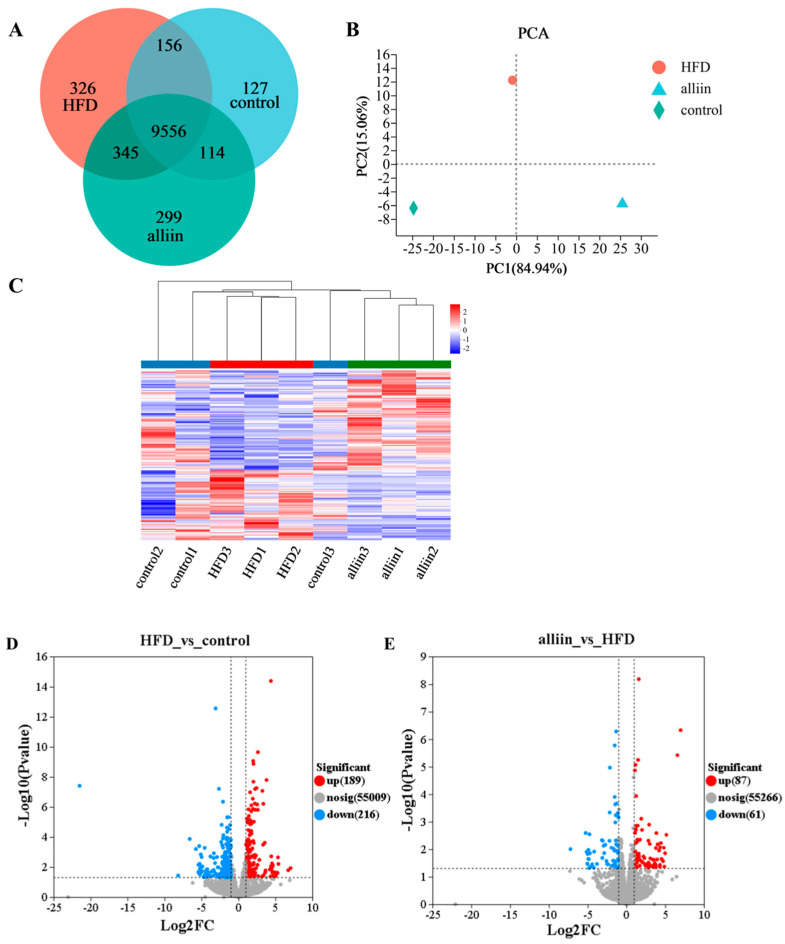
Veen plot (**A**); PCA plot (**B**); clustering heat map (**C**); and volcano plot (**D**,**E**). Red dots indicate significantly up-regulated genes, green dots indicate significantly down-regulated genes, and gray dots indicate non-significant genes. Control: the control group; HFD: the high-fat group; and alliin: the high-alliin group (120 mg/kg).

**Figure 6 foods-12-03407-f006:**
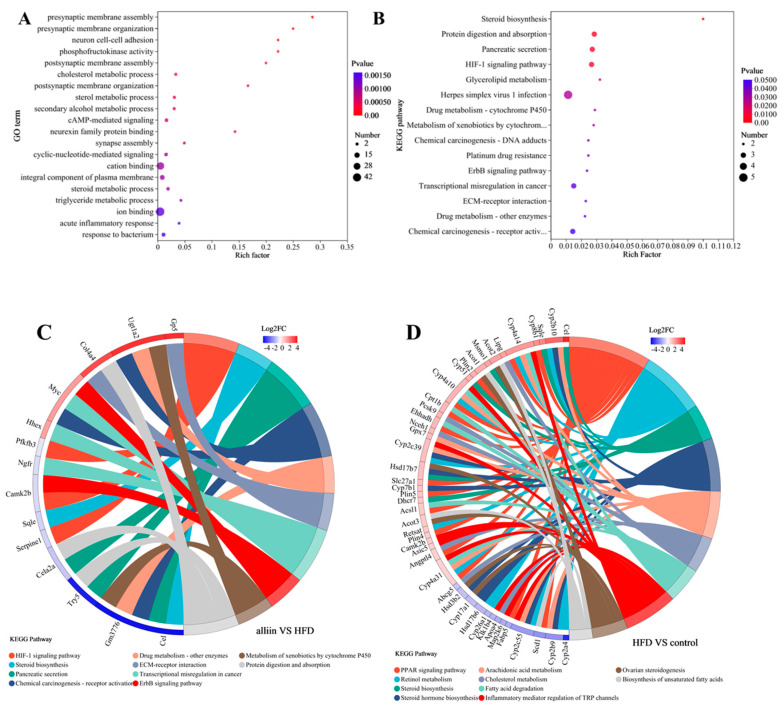
Top 20 GO enrichment (**A**): the vertical axis indicates the GO term, and the horizontal axis indicates the Rich factor; the Top 20 KEGG pathway enrichment (**B**): the vertical axis indicates the KEGG pathway name, and the horizontal axis indicates the Rich factor. The sizes of the dots indicates the number of genes in this GO term, and the colors of the dots correspond to different *p*-value ranges. Enriched chord diagram of KEGG pathway (**C**,**D**). The right side shows the Term/Pathway information on the significant enrichment of differential genes. The left side shows the genes contained in that Term/Pathway, arranged in the order of log2FC from largest to smallest.

**Figure 7 foods-12-03407-f007:**
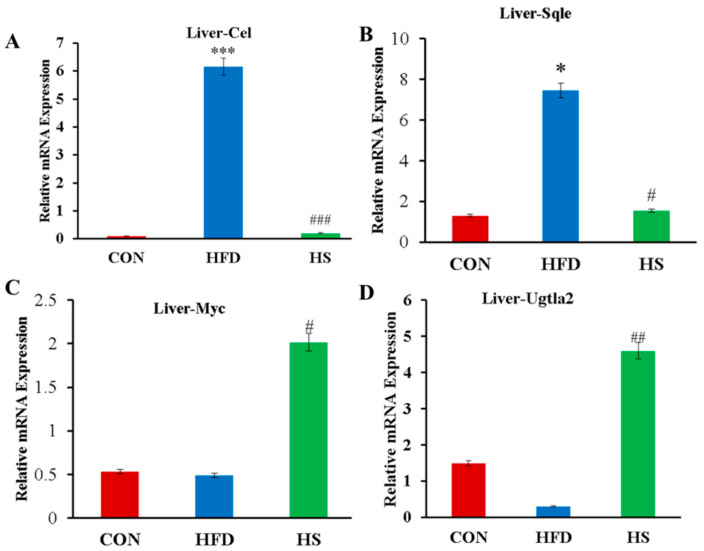
The expression levels of *Cel*, *Sqle*, *Myc,* and *Ugt1a2* in the liver. The horizontal axis indicates group, and the vertical axis indicates relevant mRNA expression. * (*p* < 0.05), and *** (*p* < 0.001) compared the CON group with the HFD group; # (*p* < 0.05), ## (*p* < 0.01) and ### (*p* < 0.001) compared the HFD group with the HS group.

**Figure 8 foods-12-03407-f008:**
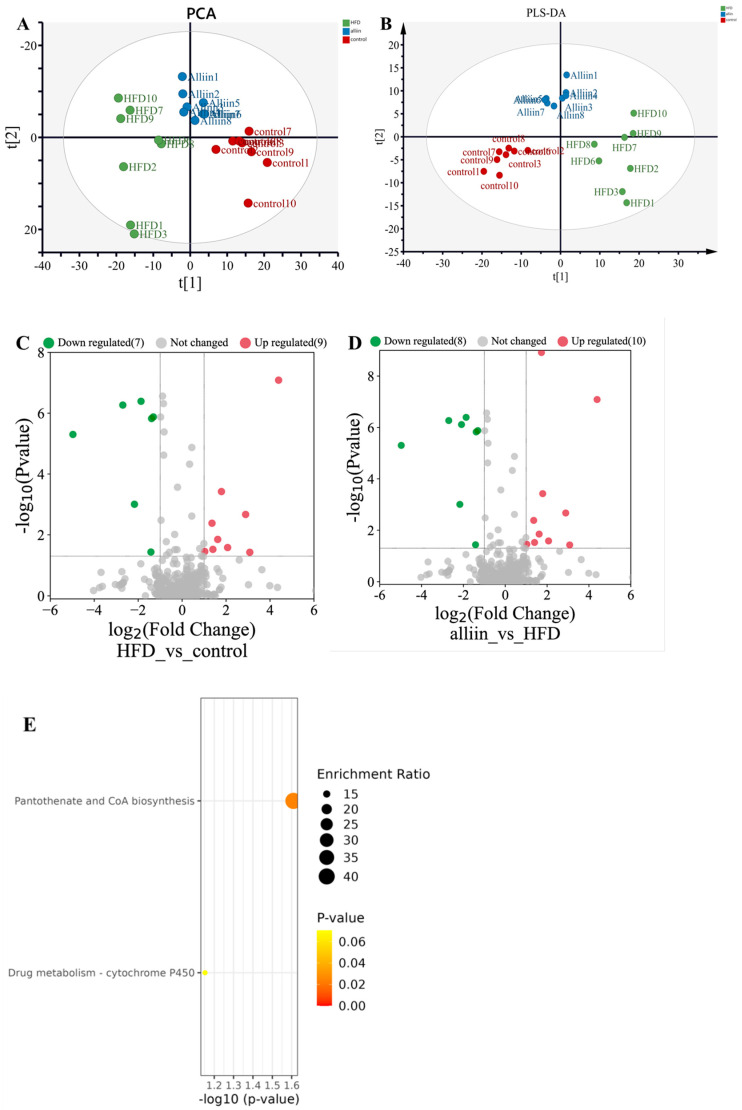
Effect of alliin on liver metabolites. PCA (**A**); PLS-DA (**B**); volcano plot (**C**,**D**); and Top 25 KEGG pathway enrichment (**E**): Vertical axis indicates the KEGG pathway name, and the horizontal axis indicates −log10 (*p*-value). Point size indicates the enrichment ratio, and the colors of the dots correspond to different *p*-value ranges. Red dots indicate significantly up-regulated genes, green dots indicate significantly down-regulated genes, and gray dots indicate non-significant genes.

**Figure 9 foods-12-03407-f009:**
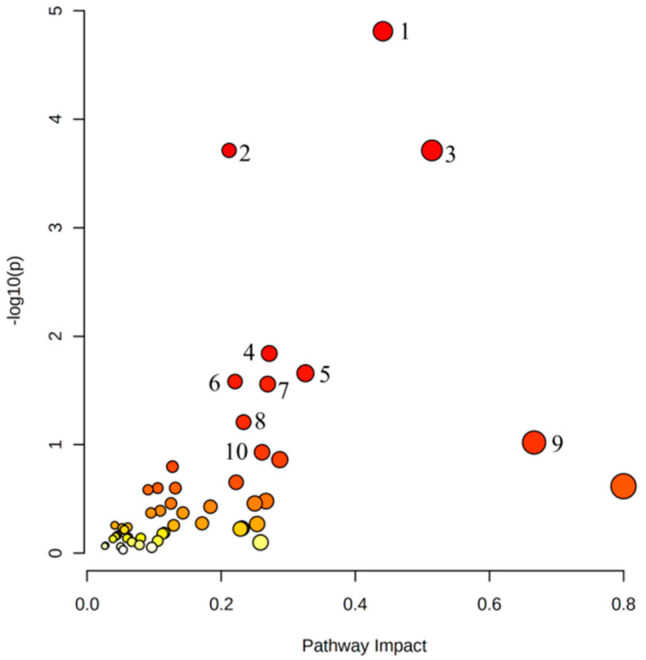
Distinctly differential Genes and metabolites pathway enrichment bubble plot. The horizontal axis is the pathway effect, and the vertical axis is the −log(*p*) value. The node color is based on the *p*-value, from light to dark, and the *p*-value from large to small; the node radius is based on its pathway impact value, from small to large, and the impact value from small to large. (1: Glycerolipid metabolism; 2: Glycerophospholipid metabolism; 3: Terpenoid backbone biosynthesis; 4: Steroid biosynthesis; 5: Retinol metabolism; 6: Drug metabolism-other enzymes; 7: Arginine biosynthesis; 8: Alanine, aspartate and glutamate metabolism; 9: Neomycin, kanamycin and gentamicin biosynthesis; and 10: Phenylalanine metabolism).

**Figure 10 foods-12-03407-f010:**
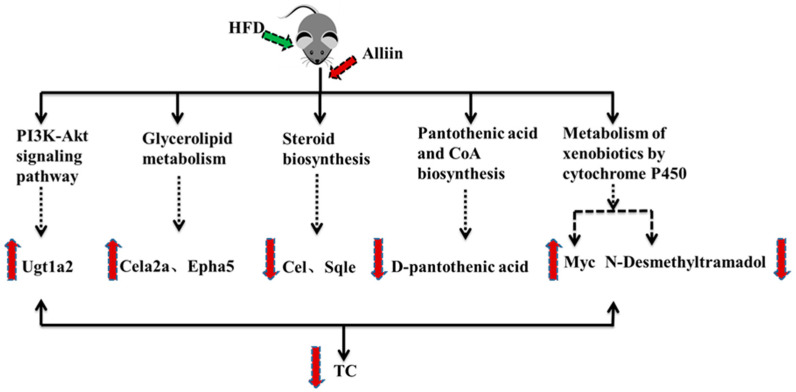
The lipid-lowering mechanism of alliin.

## Data Availability

The data presented in this study are available in the article or Appendix A.

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
