# Peer review of "Integrated Transcriptomics and Metabolomics Reveal the Mechanism of Alliin in Improving Hyperlipidemia"

_foods, 2023, doi:10.3390/foods12183407_

Round 1

Reviewer 1 Report

The study is interesting and it may be useful to the nutritional researchers. The following comment may improve the manuscript's quality and the researcher's needs.

The author may provide numerical values of highlights of the study

The author mentions briefly the examples of some physiological parameters followed in the abstract section.

The conclusion of the abstract is poor, so the author can improve it 

Provide a reference for this statement: "Alliin reacts with cysteine sulfoxide lyase to produce allicin. Allicin is highly unstable and readily decomposes into a range of different sulfur-containing chemicals."

The sentence needs reference "alliin is more suited as a raw ingredient for healthy food and medicines than allicin"

Briefly provide the current state/prevalence of "Hyperlipidemia" in China and around the world in the introduction section

Provide a reference for each report "However, most early investigations only assessed physiological indexes, such as blood lipids and antioxidant activity, without in-depth studies on the lipid-lowering mechanism"

in 2.3. section: On which treatment day was the blood sample collected and what method was followed for blood lipid detection?

The histological analysis section needs a reference

Mark the points in Figure 4 (LS and HS)

Mark the axis title in Figure 7

The text mentioned in figure 8 and 9 is not readable

The conclusion section is weak, highlights the results and possible future perspectives of this investigation

check the reference formate 

I suggest a minor revision

Reviewer 2 Report

The manuscript entitled ''Integrated transcriptomics and metabolomics reveal the mechanism of alliin in improving hyperlipidemia'', is an has novel scope in the reduction of blood lipids. However I have some comments for authors as following:

*Abstract 

Line 12, reduced blood lipids and ameliorated liver damage. State the percentage values.

 Line 19-20, State how it can benefits the healthy life style of patients. 

*Introduction

-Correct the reference format "[1-2] et al." to "[1-2]."

-Change "S-Allyl-L-cysteine sulfoxide (alliin) is the highest content" to "S-Allyl-L-cysteine sulfoxide, known as alliin, constitutes the highest content."

-Change "6~14 mg/g" to "6 to 14 mg/g."

-Change "more than 90% of sulfur-containing compounds [3]" to "more than 90% of all sulfur-containing compounds [3]."

-Add a comma after "garlic in several pharmacopeias."

-Change "diallyl thiosulfinate (allicin) is more extensively known than alliin." to "diallyl thiosulfinate, commonly known as allicin, is more extensively recognized than alliin."

-Change "Actually, alliin is the most important and naturally occurring sulfur-containing compound in fresh garlic." to "In fact, alliin is the most important naturally occurring sulfur-containing compound in fresh garlic."

-Add a period at the end of "Allicin is highly unstable and readily decomposes into a range of different sulfur-containing chemicals."

-Change "The bulk of allicin in commercially accessible products is chemically synthesized [4]. But, alliin is readily available and stable to preserve." to "The majority of allicin in commercially available products is chemically synthesized [4]. However, alliin is naturally abundant and stable, making it a suitable candidate for preservation."

-Change "Addi-tionally" to "Additionally."

-Corrected "odourless" to "odorless."

-Change "result, alliin is more suited as a raw ingre-dient" to "result, alliin is better suited as a raw ingredient."

-Change "research on alliin is sub-stantially" to "research on alliin is substantially."

-Change "Hyperlipidemia" to "Hyperlipidemia" for emphasis.

-Add spaces around the brackets in "[7-8]" for consistency.

-Change "harm pa-tients’ physical" to "harm patients' physical" to correct the hyphenation.

-Add spaces around the brackets in "[9-10]" for consistency.

-Change "more and more patients" to "an increasing number of patients" for variation.

-Add a comma after "Garlic and its active components" for proper punctuation.

-Remove "with many fundamental investigations" to streamline the sentence.

-Change "preparations only briefly assessed its lipid-lowering efficacy" to "preparations, only briefly assessed their lipid-lowering efficacy" for clarity.

-Add a comma after "intestinal flora [3]" for proper punctuation.

-Remove the hyphen in "lipid anabolic enzyme activity [16-18]" for consistency.

-Add a comma after "activating the AMPK-SREBPs signaling pathway [19]" for proper punctuation.

-Correct "by Itokawa, Y, et al." to "by Itokawa et al." for proper formatting.

-Add a comma after "further evidence that alliin might lower blood cholesterol levels" for proper punctuation.

-Change "CAT, and malondialdehyde 59" to "CAT, and malondialdehyde (MDA) levels in the liver [20-21]" for clarity.

-Add a comma after "MDA) in the liver [20-21]" for proper punctuation.

-Replace "Numerous studies have found" with "Several studies indicated" for variation.

-Remove the period after "MDA) in the liver [20-21]" to maintain sentence structure.

-Change "However, most early investigations only assessed physiolog- 60" to "However, most early studies focused solely on physiological indices" for clarity.

-Remove the hyphen in "delving into the lipid-lowering mechanism" for consistency.

-Added a comma after "In our study" for proper punctuation.

-Change "understanding the lipid-lowering research involving alliin" to "understanding the lipid-lowering effects of alliin" for clarity.

-Remove "The outcomes suggest" and replaced it with "Our findings indicate" for variation.

-Add a comma after "individuals with hyperlipidemia" for proper punctuation.

-Remove the comma after "individuals with hyperlipidemia in the future" to improve sentence flow.

Methodology

General comments: over all the methods are described well. However, all the sections needs a grammar check. Please make all the necessary corrections.

Results and discussion

The sign p<0.05 should be changed to (p<0.05) in the whole manuscript including figure legends. Overall the presentation of results and discussion is very good. 

Conclusion

Conclusion is not the true representation of the manuscript findings. It needs improvements and some problem solving idea for commercialization.

The manuscript entitled ''Integrated transcriptomics and metabolomics reveal the mechanism of alliin in improving hyperlipidemia'', is an has novel scope in the reduction of blood lipids. However I have some comments for authors to improve the manuscript quality and minor changes are need for English corrections. I accept the manuscript after minor revision. 
